# Cat and dog owners' expectations and attitudes towards advanced veterinary care (AVC) in the UK, Austria and Denmark

Sandra A. Corr[1]*, Thomas Bøker Lund[2], Peter Sandøe[3], Svenja Springer[4]

1 Division of Small Animal Clinical Sciences, School of Biodiversity, One Health and Veterinary Medicine, University of Glasgow, Glasgow, Scotland, 2 Department of Food and Resource Economics, University of Copenhagen, Frederiksberg C, Denmark, 3 Department of Veterinary and Animal Science, Department of Food and Resource Economics, University of Copenhagen, Frederiksberg C, Denmark, 4 Messerli Research Institute, Department of Interdisciplinary Life Sciences, University of Veterinary Medicine, Vienna, Austria

* sandra.corr@glasgow.ac.uk

**Data Availability Statement:** All relevant data are within the manuscript and its Supporting Information files.

## Abstract

Modern veterinary medicine offers a level of care to cats and dogs similar to that available to their owners, including blood transfusions, chemotherapy and MRI scans. The potential benefits to the animals of owners who can afford such care are obvious, but there can also be negative consequences if owners with strong emotional attachments to their pets pursue treatments that significantly reduce the quality of the animal's life while attempting to prolong it. Moreover, caring for a chronically or seriously ill animal can lead to emotional distress and financial and practical challenges for the pet owner. A questionnaire was used to survey cat and dog owners from representative samples of citizens in the UK, Austria and Denmark, to investigate owners' expectations and attitudes towards advanced veterinary care, and the factors that might influence those views. Overall, 58.4% of the pet owners surveyed believed that their pets should have access to the same *treatment options* as humans, while 51.5% believed that they should have access to the same *diagnostic tests* as humans. Owners were most likely to be neutral on the question of whether advanced veterinary care has 'gone too far' (45.3%), and to disagree with the statement that advanced care is 'unnecessary' (40.1%). In all three countries, the level of attachment owners had to their pets was most strongly associated with attitudes towards advanced care, with owners scoring higher on Lexington Attachment to Pets Scale (LAPS) being more likely to expect advanced care to be available. Other factors such as owner age, living situation (alone or not), income or possession of pet insurance were less consistently with owner attitudes. Our findings will help inform veterinarians and other health care providers about pet owner expectations and attitudes towards advanced veterinary care, and contribute to the debate on increasing specialisation within the profession.

## Introduction

Most pet owners will at some stage seek veterinary care for their animals. In doing so, they will interact with a profession that has undergone significant changes over the last two decades,

**Funding:** The study was supported via the Danish Centre for Companion Animal Welfare, of which Peter Sandøe is a leader. No grant number associated with funding The Centre gets its core funding from the charitable foundation Skibsreder Per Henriksen, R. og Hustrus Fond. https://skibsrederperhenriksensfond.dk/ The funders had no role in study design, data collection and analysis, decision to publish, or preparation of the manuscript.

**Competing interests:** The authors have declared that no competing interests exist.

both in terms of advancements in the standard of care available for companion animals, and in the underlying structure of the profession delivering that care [1, 2].

The definition of advanced veterinary care (AVC) is constantly evolving, but it sits at one end of a spectrum of appropriate care, as all veterinary treatment must meet the standard required of, and practiced by, the average reasonably prudent and competent veterinarian [2]. As discussed by others [3], 'basic' care usually involves lower skills, is less resource dependent and costs less, whereas AVC is characterised by higher costs, advanced skills, state-of-the-art techniques and equipment, but is not necessarily better. For example, a dog with painful osteo-arthritis of the hip should initially receive conservative management (e.g. controlled exercise and good analgesia) and only if the dog fails to improve should potential AVC such as a hip replacement be considered.

As a result of AVC, many animals that would previously have suffered, died or been eutha-nised due to diseases such as diabetes, renal disease or cancer can now be successfully managed in primary care practice by veterinarians using a wide range of drugs and life-saving procedures such as dialysis, blood transfusions and chemotherapy. Veterinary patients with more complex diseases or requiring advanced investigations or treatments can, in the same way as humans, be referred by their primary care veterinarian to colleagues who are specialists in fields such as orthopaedics, oncology and critical care, for advanced imaging (CT, MRI) or treatments such as radiation therapy, joint or heart valve replacements. While the availability of such care brings obvious benefits, there are also challenges: AVC is only available to pets whose owners can afford it, potentially creating moral and financial stresses for owners with financial limitations. The increasing number of choices and complexity of treatment decisions that owners are being asked to make can also be potentially overwhelming. As a broader principle, concern has also been raised over whether the prolongation of life (animal or human) *at all costs* is necessarily always in line with what may be considered the best interests of the patient, especially when the (animal) patients cannot express their own interests [2, 4]. For example, some may argue that it is not in a dog's best interest to undergo chemotherapy involving repeated veterinary visits and hospital stays, and associated malaise, to gain an extra six months with its owner.

It is within this setting that this paper explores the expectations and attitudes of dog and cat owners towards the AVC available in modern small animal practice. It is important to also consider the environment in which that AVC is delivered, and the interests of the other key stakeholders; the veterinarian delivering the care, and the practice owners and managers, who, increasingly, are not veterinarians themselves [5]. While ultimately the pet owner will decide how their pet is treated, the influence of these other stakeholders is significant, and interests can sometimes be competing.

Most veterinarians desire to offer the best care to their patients, and derive professional sat-isfaction from the development of new skills, while earning a decent living [2, 6, 7]. For exam-ple, results of an Austrian focus group study among small animal veterinarians showed that they are highly motivated by the implementation and use of new technologies that improve patient care, irrespective of whether they work as primary care or specialized veterinarians [6]. Further, recently a specific decision-ethics orientation has been identified particularly among younger and less experienced veterinarians that reflects a strong sense of responsibility to pro-vide optimal patient care and contribute to the advancement of veterinary medicine [8]. Veter-inarians who are motivated to advance veterinary care can contribute by recruiting their patients into clinical research studies on specific diseases or that assess post-operative out-comes, if the pet owner gives consent. Highly motivated veterinarians may undertake several years of postgraduate training to obtain specialist qualifications awarded by one of the recog-nized veterinary specialist Colleges in Europe (see https://ebvs.eu/about/specialist-education), or in the USA (see https://www.avma.org/education/veterinary-specialties).

In parallel, there has been a significant change in the structure of practices in many countries in response to the growing demand for high-quality veterinary services [9]. A pan-European survey in 2018 [10] found that most small animal practices (70%) were small, with fewer than five staff members, but noted a trend towards increasing corporatisation and the creation of larger practice groups. Corporatisation is rapidly increasing globally–for example in Canada, corporates are thought to control over 20% of the veterinary practices, and 'probably' 40% of veterinarians [11]. In the USA, it is estimated that 75% of specialty veterinary practices are now corporate-owned, with $45 billion worth of private equity deals being done for veterinary practices and companies between 2017–2022, resulting in companies such as Mars Veterinary Health owning over 2000 practices [12].

Corporate practice ownership is usually associated with significant investment which clearly has many potential benefits to patient care, however the businesses must generate substantial income to pay for staff salaries and the purchase and maintenance of expensive equipment, while returning profits to the investors. Significantly increased veterinary care costs, driven in part by the increasing demand for AVC, has a direct knock-on effect on increasing pet health insurance premiums in general [13, 14], potentially impacting many people who rely on pet insurance to be able to afford even basic veterinary care for their animals.

It is within this environment that dog and cat owners have to make decisions about the veterinary care for their pet(s). Without doubt, advancements in veterinary care are also driven by owner expectations [6], as they are only sustainable if owners are able and willing to pay for them, yet owners' expectations of, and attitudes towards, modern advanced veterinary care have not been fully explored. Whether dog and cat owners expect their pets to have access to the same standard of healthcare as human patients, or, conversely, think modern advanced veterinary care for animals is unnecessary or goes 'too far', was the focus of this study.

We predict that owner attitudes towards advanced care for their pet(s) will be influenced by many factors, including the relationship that they have with their animal(s). In most cases, dogs and cats kept primarily as pets have a role in the emotional lives of their owners, with the individual animal being considered irreplaceable, and often described as a 'friend' or 'family member' [1, 15]. The relationship between the owner and their pet has been evaluated using various measures of attachement, the most widely utilised being the Lexington Attachment to Pets Scale (LAPS) [16, 17]. As the benefits derived from a human-animal interaction are thought to be related to the type and depth of emotional connection between the two [18], it is not surprising that the level of attachment can also influence the decisions that owners make about their pets' healthcare [19–22].

Although the potential benefits of advanced care for the animal and the owner who cares about it are obvious, a close bond can also negatively impact the wellbeing of both. Pursuing a treatment that has a limited chance of success and significantly reduces the animal's quality of life in an effort to prolong it can seriously compromise the animal's welfare [1, 23]. In the same way, the phenomenon of caregiver burden–the emotional distress and financial and practical challenges of caring for a patient–has been recognised in the owners of seriously or chronically ill animals [24]. Other stakeholders may also be impacted, for example clients wishing to pursue treatment despite poor animal welfare has been highlighted as the most stressful ethical dilemma for veterinarians in several studies [25, 26].

This prompts the question over whether modern veterinary care can "go too far", if, for example, animal welfare becomes secondary to other concerns such as the emotional needs of the owner [27]. Although this has been the subject of much recent debate amongst animal welfare scientists, ethicists and veterinarians [2, 26, 28], opinions on AVC in this context have not been sought directly from pet owners themselves.

Although many studies exist on decision-making in veterinary treatment, the majority explore it from the perspective of the veterinarian [8, 29], or in relation to specific diseases or end of life care [30, 31] or shared decision-making between the veterinarian and owner [32–34]. However, to the best of the authors' knowledge, no studies have specifically explored pet owner expectations of and attitudes towards modern AVC in general, and little is known about the factors that might predict whether certain owners are more likely to hold certain opinions, or indeed be willing to enrol their own pet(s) into clinical research projects to help advance that care.

This study addressed the following research questions:

i.  Which diagnostic tools and treatment options do dog and cat owners expect to be available at their 'usual' veterinary practice, and how many have taken their pet to a specialist?

ii.  What attitudes do dog and cat owners have towards the AVC available in modern small animal practice?

iii.  Are owners who expect AVC to be available for their animal willing to participate by enrolling their own pet(s) into a clinical research study?

iv.  How are these attitudes influenced by factors such as pet species (dog or cat), owner age, gender, living situation, income, possession of pet health insurance, and level of emotional attachment to their pet?

Differences and similarities between owners in three countries, the UK, Austria and Denmark, were explored. These countries were selected to represent diversity in the veterinary sector, as smaller independently owned practices still predominate in Austria, whereas larger practices are more common in Denmark, and in the UK large practices owned by corporations are becoming dominant [5, 10].

Our study aims to provide veterinarians, practice owners and others related to the profession, such as insurance companies, with insights into pet owners expectations and attitudes towards AVC, and the factors that might predict those expectations and attitudes.

## Materials and methods

Ethical approval for the entire project was obtained from the Research Ethics Committee of SCIENCE and HEALTH (ReF: 504-010300/22-5000) at the University of Copenhagen.

### Survey design and measurements

The questionnaire consisted of three sections (see S1 File). For the following detailed description, only those questions relevant to the present paper, in Sections A and B, are considered here.

Section A included 19 closed-ended questions on respondent socio-demographics, pet ownership (dog and/or cat) and insurance, and level of emotional attachment, as well as veterinary practice aspects (attendance, availability of equipment) and pet health insurance status ("yes", "no", "not anymore"). Further, the 23 items that are used in the Lexington Attachment to Pets Scale (LAPS) [16, 17] to measure owners' attachment to their animal were included in the section. The LAPS includes statements such as 'I believe my pet is my best friend', 'I believe that pets should have the same rights and privileges as family members' and 'Pets deserve as much respect as humans do'. Response options were 1 'strongly disagree', 2 'somewhat disagree', 3 'somewhat agree' and 4 'strongly agree'. If respondents had both cat(s) and dog(s), they were asked which species their favorite pet was, and instructed to think about that favorite pet when responding to the 23 LAPS statements.

The section also inquired about which of four basic (radiography, ultrasound, dental equipment, in-house laboratory) and four advanced (CT scanner, MRI, arthroscopy, endoscopy) diagnostic and treatment options owners would expect to be available at the pracice they usually attend. Short explanations of each were presented, to ensure that respondents understood how the equipment was used in practice, and a multiple response format was used, so that respondents were able to indicate all the options they would expect to be available. The answer options 'none of the above' and 'I don't know' were also provided. Based on their answers, respondents were divided into two groups: the 'basic only' group consisted of owners who selected only one or more basic options, while the 'advanced' group consisted of owners who, along with basic options, had also selected one or more advanced options (see Table 3). Further, respondents were asked whether they have ever taken their dog(s) or cat(s) to specialist veterinarian. Available answer options were 'yes', 'no' and 'I don't know'.

<u>Section B</u> contained a matrix to measure owners' attitudes towards advances in small animal practice, based on their level of agreement with seven statements including, for example, 'My pet should have access to the same diagnostic tests that are available to human patients', 'I would enrol my pet in a research study to help advance veterinary care, as long as the risk of potential complications was low', or 'The advanced care available in modern veterinary medicine has gone 'too far', putting animals through 'too much". Respondents were able to indicate their level of agreement by using a 7-point Likert scale from 1 'strongly disagree' up to 7 'strongly agree'.

At the end of the questionnaire, respondents were asked about their total gross household income per year, within specified ranges, and also given the option to decline to say.

## Survey development

The questionnaire was developed with a view to comparing themes and results from a previous transnational study (involving the same three countries) on veterinarians' decision-making in the context of modern veterinary practice [8]. Informed by literature review, items were developed to consider aspects related to the relationship between people and their pets and how this might influence their expectations and attitudes towards veterinary care. Two stages of pretesting were carried out: initially, 15 cognitive interviews [35, 36] were conducted with five (dog and /or cat) owners in each country to check whether the content was clear or could be misunderstood; where appropriate, questions were subsequently either reformulated, or in some cases, removed. Next, an online pre-test phase was conducted with 123 Austrian, 152 Danish and 117 UK citizens, including 34 dog and 39 cat owners in Austria, 30 dog and 24 cat owners in Denmark and 41 dog and 40 cat owners in the UK. Relevant comments that were likely to improve data quality were incorporated into the final version of the questionnaire in all three languages.

## Recruitment of participants and representativity of the sample

The present study forms a part of a larger body of work on dog and cat owners' attitudes towards different aspects of modern small animal practice. Participants for the study were recruited by NORSTAT, a European-based survey company, to gain a representative sample of UK, Danish and Austrian citizens (over 17 years of age), including pet owners. Norstat manages citizen panels in several European countries where the members of these panels have agreed to receive invitations to participate in research. In Denmark, panel members are recruited through a mixture of telephone interviews (random digit dialing), and online sources (typically internet pages). In the UK and in Austria, panel members are only recruited through online sources. The invitees were recruited from NORSTAT's pre-established citizen panel in

the three study countries. In each country, Norstat conducted a random draw of panel members. During data collection, Norstat continuously monitored the number of respondents that had completed the questionnaire across different age groups, geographical location, and gender. If particular groups were under-represented (compared with the countries' census), extra invitations were issued within this group (still using the random draw principle within the group).

Invitations to participate were issued to 17,747 panel members (5207, 6075 and 6465 in Austria, Denmark and the UK, respectively) via an e-mail containing a link to the online questionnaire. The invitation provided background information on the study, including the participating Universities, ethical approval, anonymity of data and participant's rights. Before participants were directed to the survey, important information was provided on the opening page to obtain their informed consent. Participants were advised that completion of the questionnaire is voluntary, they can exit at any point prior to submitting the final answers, that responses will be passed to the researchers in an anonymized form and no information can be traced back to them. Since this was an online survey, participants provided consent by clicking the 'Next' button at the bottom of the consent form, also confirming that they were over 17 years old, in order to proceed to the questionnaire. The survey was open from 11-25th March 2022 in Austria, 11-24th March 2022 in Denmark, and 8-23rd March 2022 in the UK.

Of the 17,747 invitations issues, the survey weblink was clicked on by 4885 individuals, and completed questionnaires were received from 4,610 individuals (1500, 1552 and 1558 from Austria, Denmark and the UK respectively), reflecting a dropout rate of 5.26%. The response rate was 30.34% for Austria, 27.49% for Denmark and 25.29% for UK. The final sample relevant to the present paper (cat and dog owners) consisted of 2117 owners: 844 dog owners, 872 cat owners and 401 owners who keep both dog(s) and cat(s). To account for demographic misrepresentation in the study data, all analyses of means and proportions (but not inferential statistics and regressions) were reported with weighted data using a weight variable that adjusts the data to the three countries' population census regarding age, gender, and region.

## Data analysis

IBM® SPSS® Statistics version 28.0 (IBM® SPSS® Statistics, Chicago, IL, USA) was used for all analyses. Univariate descriptive statistics were presented in tables or text. For bivariate analysis, Chi-Square were conducted to test whether the distribution between dogs and cat owners or owners who owned both species differed. Further, Chi-Square and Kruskal-Wallis *H* tests were conducted to test whether the frequency distribution differed between the Austrian, Danish and UK sub-populations. The significance level was 0.05. Results of univariate analyses were conducted with weighted data to take into account the effects of any sample imbalances and to bring them more in line with the of dog and cat owners in the three countries. Bonferroni correction was applied for all multiple comparisons and significant variables.

Further, four ordinal regression analyses were conducted separately for each country to examine what determines owners' attitudes towards advances in small animal practice. A main explanatory variable was animal species (i.e. whether the owner has a cat or a dog). This dichotomous variable was constructed using a question where respondents stated whether they had one or more cats or dogs: 401 pet owners (18.6%) reported having both species. Those owners with both were then asked to think about their favourite pet: of the 401 owners, 263 (65.6%) chose a dog and 107 (26.7%) chose a cat as their favourite pet. Those who chose a cat were ascribed to be cat owners, and vice versa if a dog was chosen. As some owners chose another species e.g. a horse as their favourite animal, the eligible sample size decreased slightly (N = 31 out of 2117 dog and cat owners; 1.5%).

In doing so, the following four statements focusing on this were treated as ordered variables and were included as dependent variables one by one: (1) 'My pet should have access to the same treatment options that are available to human patients.', (2) 'I would enrol my pet in a research study to help advance veterinary care, as long as the risk of potential complications was low.', (3) 'The advanced care available in modern veterinary medicine has gone 'too far', putting animals through 'too much.'' and (4) 'The advanced care available in modern veterinary medicine is unnecessary—animals should not be treated in the same way as humans.' Owner's age (range: AT: 18–81 years; DK: 18–86 years, UK: 18–83 years) and their emotional attachment to their animal (mean score of LAPS) were included as continuous predictor variables. Gender (1 = male, 2 = female), animal species (1 = dog, 2 = cat), existence of health insurance (1 = yes, 2 = no/not anymore) and total gross household income per year (0 = 'I don't know/'Prefer not to say'; 1 = low; 2 = middle; 3 = high;) and owner living status (1 = live alone; 2 = live not alone) were inserted as categorical variables. For each regression analysis, we evaluated the overall regression model fit using the LR chi2 test and Nagelkerke's Pseudo-R-Square (see S2 File).

Finally, to explore whether owners who believe that their pet should have access to advanced treatment consider that both their veterinarian, and their own pet, should contribute to the development of that care, Pearson correlation analyses were conducted on the responses to the following statements: (1) 'My vet should offer my pet the most advanced treatment that is available', (2) 'It is important that veterinarian contributes knowledge to the advancement of veterinary care for future patients.' and (3) 'I would enrol my pet in a research study to help advance veterinary care, as long as the risk of potential complications was low.'

## Results

### A. Dog and cat owner socio-demographics

As shown in Table 1, a similar number of respondents from Denmark and the UK had dogs or cats, however there were significantly fewer Austrian dog owners, and significantly more Austrian cat owners, than in both Denmark and the UK (p<0.001). Significantly fewer respondents from Denmark had both cat(s) and dog(s) compared to those from the UK and Austria.

Table 2 highlights socio-demographic differences in cat and dog ownership across countries. The age of respondents was not significantly different between the UK and Austria, but fewer Danish respondents were in the youngest age range ($p_{AT}$ = 0.009; $p_{UK}$<0.001). Gender balance was similar between countries, and the majority of people in all three countries lived with others (76.9–80.6%) rather than living alone. Only a small proportion of respondents had a veterinary background. Respondent income showed a similar distribution between low,

**Table 1. Number (N,n) and proportions (%) of respondents owning either dog(s), cat(s) or both listed for all countries (N = 2117) and separately for Austria (n = 800), Denmark (n = 626) and the UK (n = 691).**

| Species | All<br>N = 2117* (%)** | Austria<br>n = 800* (%)** | Denmark<br>n = 626* (%)** | UK<br>n = 691* (%)** | Test* |
|---|---|---|---|---|---|
| **Dog(s)** | 844 (40.4) | 225 (28.6) | 308 (49.7) | 311 (45.3) | **Cats vs. Dogs:**<br>Chi²(2) = 61.619, p<0.001<br>AT vs. DK: p<0.001<br>AT vs. UK: p<0.001<br>DK vs. UK: p = 0.909 |
| **Cat(s)** | 872 (41.0) | 391 (49.1) | 241 (38.5) | 240 (34.2) | |
| **Both (dog(s) and cat(s))** | 401 (18.6) | 184 (22.3) | 77 (11.9) | 140 (20.5) | |

*Number of respondents (N,n) and analyses with inferential statistics were calculated with unweighted data

**Proportions were calculated with weighted data; rounding errors lead to some differences between rounded-off numerical values and actual values

**Table 2. Pet owner socio-demographic information including age, age groups, gender, living status, work in the veterinary field, presence of pet health insurance and information on income listed for all countries (N = 2117) and separately for Austria (n = 800), Denmark (n = 626) and the UK (n = 691).**

| | All (N = 2117) | Austria (n = 800) | Denmark (n = 626) | UK (n = 691) | Test* |
|---|---|---|---|---|---|
| **Age (years)** | | | | | |
| Mean ± Std** | 46.26±16.36 | 45.90±16.14 | 47.86±16.25 | 45.26±16.62 | |
| Median [IQR]** | 47.0 [32;59] | 47.0 [32;59] | 48 [35;59] | 45 [30;58] | |
| **Age Groups (years)** | N* (%)** | n* (%)** | n* (%)** | n* (%)** | |
| 18–30 | 518 (21.9) | 227 (22.4) | 111 (17.4) | 180 (25.3) | H(2) = 25.670, p<0.001 |
| 31–40 | 354 (17.0) | 143 (16.6) | 94 (15.8) | 117 (18.4) | AT vs DK: p = 0.009 |
| 41–50 | 405 (18.4) | 142 (16.6) | 145 (22.3) | 118 (16.9) | AT vs UK: p = 0.120 |
| 61–60 | 412 (20.7) | 160 (22.5) | 128 (20.9) | 124 (18.5) | DK vs UK: p<0.001 |
| >60 | 427 (22.0) | 128 (21.9) | 147 (23.6) | 152 (20.8) | |
| Prefer not to say | 1 (<0.1) | 0 (0.0) | 1 (0.1) | 0 (0.0) | |
| **Gender** | | | | | |
| Male | 853 (46.0) | 314 (45.7) | 247 (44.3) | 292 (47.7) | $\chi^2(2)$ = 1.690, p = 0.430 |
| Female | 1256 (53.6) | 482 (53.8) | 378 (55.5) | 396 (51.8) | |
| Neither of these | 8 (0.4) | 4 (0.5) | 1 (0.2) | 3 (0.4) | |
| **Do you live alone?** | | | | | |
| Yes | 444 (21.3) | 175 (23.1) | 136 (21.2) | 133 (19.4) | $\chi^2(2)$ = 1.848, p = 0.397 |
| No | 1673 (78.7) | 625 (76.9) | 490 (78.8) | 558 (80.6) | |
| **Do you work in the veterinary field?** | | | | | |
| Yes | 138 (6.8) | 57 (6.8) | 11 (1.8) | 70 (11.2) | |
| No | 1979 (93.2) | 743 (93.2) | 615 (98.2) | 621 (88.8) | |
| **Do you have pet health insurance?** | | | | | |
| Yes | 876 (41.7) | 171 (21.3) | 350 (56.3) | 355 (51.5) | $\chi^2(2)$ = 214.933, p<0.001 |
| No/not anymore | 1241 (58.3) | 629 (78.7) | 276 (43.7) | 336 (48.5) | AT vs DK: p<0.001 |
| | | | | | AT vs UK:p<0.001 |
| | | | | | DK vs UK: p = 0.078 |
| **Respondent Income** | | | | | |
| Low+ | | 220 (26.6) | 138 (21.3) | 187 (26.3) | H(2) = 15.410, p<0.001 |
| Middle++ | | 247 (31.1) | 175 (28.3) | 224 (32.0) | AT vs. DK: p<0.001 |
| High+++ | | 197 (25.3) | 219 (35.4) | 210 (32.0) | AT vs UK: p = 0.112 |
| | | | | | DK vs UK: p = 0.297 |
| Don't know/ prefer not to say | | 136 (17.0) | 94 (14.9) | 70 (9.7) | |

*Number of respondents (N,n) and analyses with inferential statistics were calculated with unweighted data

**Proportions, mean±std and median [IQR] were calculated with weighted data; rounding errors lead to some differences between rounded-off numerical values and actual values

**Low+** Austria (Euro) < 13450–26899; Denmark (DK) < 100.000–300.000; UK £<11,200–22,399

**Middle++** Austria (Euro) 26900–53799; Denmark (DK) 300.001–600.000; UK £22,400–44,799

**High+++** Austria (Euro) 53800–134500; Denmark (DK) 600.001–1.000.001; UK £44,800–112,000

middle and high categories both within and across countries, although there were significantly more high-income owners in Denmark than Austria (p<0.001). However, significantly fewer Austrian respondents had pet insurance (21.3%, p<0.001) compared to owners in the UK (51.5%) and Denmark (56.3%).

## B. Veterinary practice related information and owner expectations

There were significant differences across the three countries in the practice types attended by the respondents. While 69–71.7% of cat and dog owners respectively in Austria, and 57.9–

**Table 3. Diagnostic and treatment options pet owners in each country (all N = 2117; Austria n = 800; Denmark n = 626; UK n = 691) expect at the practice they _usually_ attend grouped in "none/I don't know", "basic only" and "advanced".**

| Diagnostic /treatment option | All countries N = 2117* (%)** | Austria n = 800* (%)** | Denmark n = 626* (%)** | UK n = 691* (%)** | Test* |
|---|---|---|---|---|---|
| None / I don't know | 496 (23.5) | 141 (17.4) | 207 (34.0) | 148 (21.2) | H(2) = 46.638, p<0.001 |
| Basic only[1] | 556 (26.1) | 268 (33.4) | 157 (25.1) | 131 (19.0) | AT vs DK: p<0.001 AT vs UK: p = 0.063 |
| Advanced[2] | 1065 (50.3) | 391 (49.2) | 262 (40.9) | 412 (59.8) | DK vs UK: p<0.001 |

*Number of respondents (N,n) and analyses with inferential statistics were calculated with unweighted data

**Proportions were calculated with weighted data; rounding errors lead to some differences between rounded-off numerical values and actual values

[1]'Basic only' options: Radiography, Ultrasound, In-house laboratory, Dental equipment

[2]'Advanced' options: Endoscopy, Arthroscopy, MRI scanner, CT scanner

58.6% of cat and dog owners respectively in Denmark attended practices with 1–3 vets, this was the case for only 37.4% of cat and 36.9% of dog owners in the UK (p<0.001). A significantly higher proportion of owners in the UK (32.5%) attended practices with four or more vets, compared to Austrian (11.8%) or Danish owners (26.2%) (p<0.001). **(S3 File).**

Overall, 50.3% of the respondents expected _advanced_ diagnostic and treatment options to be available at their usual practice, while 23.5% either did not know, or expected none to be available **(Table 3)**. Owners in the UK were significantly more likely to expect advanced options to be available at their usual practice than Austrian owners (p = 0.063), while Danish owners were significantly less likely to expect them than UK (p<0.001) or Austrian owners (p<0.001).

Overall, 18.3% of dog owners and 10.3% of cat owners have taken their pet to a specialist **(Table 4)**. Significantly fewer dogs (8.9%) or cats (5.6%) had been taken to see a specialist in Denmark compared to Austria ($p_{dog,cat}$<0.001) or the UK ($p_{dog}$<0.001, $p_{cat}$ = 0.009).

## C. Pet owner attitudes towards advanced veterinary care and willingness to enrol pet into clinical research

Overall, more than half of pet owners agreed that their pet should be able to **_access the same diagnostic tests_** (50.6%) and **_treatment options_** (58.4%) as human patients, and that their **_vets should offer the most advanced care_** (63.6%) (**Table 5**). While the majority also agreed that **_vets had an important role_** in contributing knowledge to advance veterinary care (64.4%), they were less likely to agree to **_enroll their own pet in a research study_** (39.0%). Owners were most likely to be neutral on the question of whether advanced care has **_'gone too far'_** (45.3%), and to disagree with the statement that it is **_'unnecessary'_** (40.1%).

However significant differences exist between owners in the three countries. Significantly more UK owners than Austrian owners agreed that their pet should have access to the same **_diagnostic tests_** as human patients (62.8% and 52.5% respectively, p<0.001); access to the same **_treatment options_** as human patients (70.9% and 64.9% respectively, p = 0.008) and that their vet should **_offer their pet the most advanced treatment_** that is available (76.6% and 69.1% respectively, p<0.001). Notably, Danish owners were significantly (p<0.001) less likely to agree with any of the three statements (34.2%, 35.8% and 41.6% respectively) than either UK or Austrian owners.

When considering whether **_advanced veterinary care has 'gone too far' putting animals through too much,_** there was no significant difference in level of owner agreement between countries (p = 0.291). Overall, 45.3% of owners were neutral, with slightly more agreeing (32.0%) than disagreeing (22.6%) with the statement. However, when asked whether **_advanced_**

**Table 4. Owner responses when asked 'Have you ever taken your pet to a Specialist?' considered by species (dog or cat), and country (all N = 2117; Austria n = 800; Denmark n = 626; UK n = 691).**

| | All<br>N = 2117 | Austria<br>n = 800 | Denmark<br>n = 626 | UK<br>n = 691 | Test* |
|---|---|---|---|---|---|
| **Dogs** | **n = 1245* (%)\*\*** | **n = 409* (%)\*\*** | **n = 385* (%)\*\*** | **n = 451* (%)\*\*** | |
| Yes | 221 (18.3) | 108 (26.2) | 35 (8.9) | 78 (19.3) | $\chi^2(2)$ = 41.413, p<0.001 |
| No | 991 (79.1) | 288 (70.9) | 338 (88.1) | 365 (78.9) | AT vs DK: p<0.001<br>AT vs UK: p<0.001<br>DK vs UK: p<0.001 |
| I don't know | 33 (2.6) | 13 (3.0) | 12 (3.1) | 8 (1.8) | |
| **Cats** | **n = 1273 (%)** | **n = 575 (%)** | **n = 318 (%)** | **n = 380 (%)** | |
| Yes | 130 (10.3) | 69 (11.7) | 17 (5.6) | 44 (12.0) | $\chi^2(2)$ = 11.832, p = 0.003 |
| No | 1107 (86.8) | 483 (84.6) | 298 (93.5) | 326 (84.6) | AT vs DK: p<0.001<br>AT vs UK: p = 0.783<br>DK vs UK: p = 0.009 |
| I don't know | 36 (2.9) | 23 (3.7) | 3 (1.0) | 10 (3.4) | |

*Number of respondents (N,n) and analyses with inferential statistics were calculated with unweighted data

**Proportions were calculated with weighted data; rounding errors lead to some differences between rounded-off numerical values and actual values

*care was unnecessary as animal should not be treated as humans,* significantly more Austrian owners (47.9%, p<0.001) *disagreed* with the statement than either UK owners (39.2%), or Danish owners (31.2%), the latter being more likely to be neutral (37.5%).

On the question of whether *vets should contribute knowledge to advance veterinary care,* the majority of owners in Austria (69.6%) and the UK (71.5%) agreed, but significantly fewer Danish owners agreed (49.5%) (p< 0.001). However, owners were in general less willing to *enrol their own pet into a research study:* 51.3% of UK owners would, in contrast to only 37.8% of Austrian and 26.6% of Danish owners, with statistically significant differences between all three countries (p<0.001). Exploring these data further using Pearson Correlation analysis, we found that the owners in all three countries who believed that their pet should have access to the same treatment options that are available to human patients were significantly more likely to agree that their vet should contribute to advancing that care (Austria: r = 0.543, p<0.01; Denmark: r = 0.490, p<0.001; UK: r = 0.478, p<0.001) and to be willing to enrol their own pet in a research study (Austria: r = 0.215, p<0.01; Denmark: r = 0.266; p<0.001; UK: r = 0.299, p<0.001). (**S4 File**).

## What might explain the owner attitudes to advances in small animal practice and willingness to enrol their pet in clinical research?

Ordinal regression analysis was undertaken to determine the extent to which specific factors—owners' age, gender, living circumstances (alone or not), animal species, attachment to the animal (LAPS), income or possession of pet health insurance—influenced responses to the four statements chosen as indicators of owner attitudes towards AVC (**Table 6**).

In all three countries, the factor most consistently associated with owner attitudes towards AVC was their level of emotional attachment to their animal (LAPS score) (**Table 6**; **S2 File**). Owners with a *higher* LAPS score were significantly more likely to agree that pets should have access to the same treatment options that are available to humans ($p_{AT,DK;UK}$<0.001), and to agree to enrol their pet into a research study, in all three countries ($p_{AT,DK;UK}$<0.001). However, while Austrian owners with a *higher* LAPS score were more likely to agree that advanced care has gone too far (p = 0.012), both Austrian and Danish owners with *lower* LAPS scores were more likely to agree that advanced care is unnecessary ($p_{AT, DK}$<0.001).

**Table 5. Pet owners' attitudes towards advanced veterinary care (AVC) in small animal veterinary practice considered by country (all N = 2117; Austria n = 800; Denmark n = 626; UK n = 691).**

| | It is important that… | Level of agreement* | All countries N = 2117* (%)** | Austria n = 800* (%)** | Denmark n = 626* (%)** | UK n = 691* (%)** | Test* |
|---|---|---|---|---|---|---|---|
| 1 | My pet should have access to the same diagnostic tests that are available to human patients | Disagreement (1–3) | 483 (23.0) | 157 (19.8) | 234 (37.7) | 92 (13.8) | **H(2) = 160.925, p<0.001** UK vs. DK: p<0.001 UK vs. AT: p<0.001 DK vs. AT: p<0.001 |
| | | Neutral (4) | 552 (26.4) | 216 (27.8) | 178 (28.1) | 158 (23.3) | |
| | | Agreement (5–7) | 1082 (50.6) | 427 (52.5) | 214 (34.2) | 441 (62.8) | |
| | | **Mean ± Std.** ** | **4.45±1.59** | **4.58 ±1.49** | **3.79 ±1.65** | **4.87 ±1.47** | |
| 2 | My pet should have access to the same treatment options that are available to human patients | Disagreement (1–3) | 357 (17.0) | 87 (11.0) | 218 (35.3) | 52 (7.8) | **H(2) = 253.496, p<0.001** UK vs. DK: p<0.001 UK vs. AT: p = 0.008 DK vs. AT: p<0.001 |
| | | Neutral (4) | 510 (24.6) | 184 (24.1) | 182 (28.9) | 144 (21.3) | |
| | | Agreement (5–7) | 1250 (58.4) | 529 (64.9) | 226 (35.8) | 495 (70.9) | |
| | | **Mean ± Std.** ** | **4.71±1.53** | **4.95 ±1.35** | **3.87 ±1.66** | **5.20 ±1.29** | |
| 3 | I would enrol my pet in a research study to help advance veterinary care, as long as the risk of potential complications was low. | Disagreement (1–3) | 673 (32.2) | 266 (34.3) | 268 (43.3) | 139 (20.2) | **H(2) = 123.456, p<0.001** UK vs. DK: p<0.001 UK vs. AT: p<0.001 DK vs. AT: p<0.001 |
| | | Neutral (4) | 614 (28.8) | 224 (27.9) | 190 (30.1) | 200 (28.5) | |
| | | Agreement (5–7) | 830 (39.0) | 310 (37.8) | 168 (26.6) | 352 (51.3) | |
| | | **Mean ± Std.** ** | **4.00±1.64** | **3.97 ±1.60** | **3.46 ±1.67** | **4.51 ±1.49** | |
| 4 | My vet should offer my pet the most advanced treatment that is available. | Disagreement (1–3) | 248 (11.6) | 62 (7.7) | 158 (25.3) | 28 (4.0) | **H(2) = 233.001, p<0.001** UK vs. DK: p<0.001 UK vs. AT: p<0.001 DK vs. AT: p<0.001 |
| | | Neutral (4) | 523 (24.8) | 182 (23.2) | 208 (33.1) | 133 (19.4) | |
| | | Agreement (5–7) | 1346 (63.6) | 556 (69.1) | 260 (41.6) | 530 (76.6) | |
| | | **Mean ± Std.** ** | **4.90±1.38** | **5.07 ±1.23** | **4.18 ±1.52** | **5.34 ±1.14** | |
| 5 | It is important that my vet contributes knowledge to the advancement of veterinary care for future patients. | Disagreement (1–3) | 168 (8.0) | 56 (6.9) | 80 (13.3) | 32 (4.0) | **H(2) = 105.439, p<0.001** UK vs. DK: p<0.001 UK vs. AT: p = 0.090 DK vs. AT: p<0.001 |
| | | Neutral (4) | 585 (27.6) | 184 (23.5) | 235 (37.2) | 166 (23.9) | |
| | | Agreement (5–7) | 1364 (64.4) | 560 (69.6) | 311 (49.5) | 493 (71.5) | |
| | | **Mean ± Std.** ** | **4.97±1.26** | **5.08 ±1.20** | **4.53 ±1.32** | **5.23 ±1.18** | |
| 6 | The advanced care available in modern veterinary medicine has gone 'too far', putting animals through 'too much' | Disagreement (1–3) | 483 (22.6) | 206 (24.8) | 122 (20.6) | 155 (22.0) | **H(2) = 2.467, p = 0.291** |
| | | Neutral (4) | 954 (45.3) | 339 (43.8) | 314 (49.6) | 301 (43.3) | |
| | | Agreement (5–7) | 680 (32.0) | 255 (31.4) | 190 (29.8) | 235 (34.7) | |
| | | **Mean ± Std.** ** | **4.13±1.29** | **4.09 ±1.27** | **4.08 ±1.24** | **4.21 ±1.35** | |

*(Continued)*

**Table 5.** (Continued)

| It is important that... | | Level of agreement* | All countries N = 2117* (%)** | Austria n = 800* (%)** | Denmark n = 626* (%)** | UK n = 691* (%)** | Test* |
|---|---|---|---|---|---|---|---|
| 7 | The advanced care available in modern veterinary medicine is unnecessary—animals should not be treated in the same way as humans. | Disagreement (1–3) | 860 (40.1) | 394 (47.9) | 194 (31.2) | 272 (39.2) | **H(2) = 51.189, p<0.001** UK vs. DK: p = 0.057 UK vs. AT: p<0.001 DK vs. AT: p<0.001 |
| | | Neutral (4) | 657 (31.5) | 222 (29.5) | 234 (37.5) | 201 (28.5) | |
| | | Agreement (5–7) | 600 (28.4) | 184 (22.6) | 198 (31.4) | 218 (32.3) | |
| | | **Mean ± Std.** ** | **3.65±1.61** | **3.34 ±1.61** | **3.90 ±1.46** | **3.76 ±1.68** | |

*Number of respondents (N,n) and analyses with inferential statistics were calculated with unweighted data

**Proportions and mean±std were calculated with weighted data; rounding errors lead to some differences between rounded-off numerical values and actual values

Owner age was also associated with willingness to enrol their pet into a research study in all three countries, with younger owners being significantly more likely to do so than older owners ($p_{AT,DK;UK}$ <0.001). Within countries, it is notable that while older Danish owners were significantly more likely to agree that advanced care has gone too far (p = 0.007) or is unnecessary (p<0.001), in the UK it is younger owners who are more likely to agree with both statements (p<0.001).

The effect of gender, income and the possession of insurance on attitudes varied across countries. Danish owners were significantly more likely to agree that their pet should have access to advanced care if they had insurance than if they did not (p = 0.002), however this was not a significant factor for UK or Austrian owners. Austrian owners with insurance were significantly more likely to agree to enroll their pet in a clinical trial (p<0.001), but they were also more likely to believe that advanced care has gone too far (p<0.001) or is unnecessary (p<0.001) compared to owners without insurance for their animal.

In the UK in particular, owner income was strongly associated with attitudes, where those on low (p = 0.042–0.016) or middle (p = 0.016 - < 0.001) incomes were in most cases significantly less likely to agree with all four statements than high income owners (p = 0.042 - <0.001). In contrast, income had no effect on Austrian owner responses, and only affected Danish owner responses to the question of whether animals should have access to advanced care, where low (p = 0.04) and middle (p = 0.006) income owners were significantly more likely to agree than high income owners. No significant relationship between income and possession of insurance was found for Austrian and Danish owners, however in the UK, more high-income owners (41.7%) than low-income owners (20.7%) had insurance (p<0.001) (**S5 File**).

In contrast, animal species and whether or not the owner lived alone were not strongly associated with attitudes in most cases; living alone was only a significant factor in the UK, where those who lived alone were more likely to agree that care had gone too far (p = 0.036) or was unnecessary (p = 0.008).

As LAPS score was strongly associated with attitudes across all countries, judged by the highly significant p-values (p<0.001) that in most cases prevailed between LAPS and the attitude/willingness measures in Table 6, potential correlations between LAPS score and age, gender, living status, possession of pet health insurance and income were tested (**S6 File**). Across all countries, female owners had significantly higher LAPS scores than male owners ($p_{ALL}$ =

**Table 6. Factors (age, gender, living alone or not, income, insurance, species and LAPS) associated with pet owners' attitudes towards advanced veterinary care (AVC) in small animal practice and willingness to enrol pet in clinical research in Austria, Denmark and the UK.**

| PREDICTORS | COUNTRY | STATEMENTS | | | |
|---|---|---|---|---|---|
| | | My pet should have access to the same treatment options that are available to human patients | I would enrol my pet in a research study to help advance veterinary care as long as the risk of potential complications was low | The advanced care available in modern veterinary medicine has gone 'too far' putting animals through 'too much' | The advanced care available in modern veterinary medicine is unnecessary–animals should not be treated in the same way as humans |
| **Age** | Austria | n/s | Younger more likely to agree p = 0.006 | n/s | n/s |
| | Denmark | | Younger more likely to agree p = 0.002 | Older more likely to agree p = 0.007 | Older more likely to agree p<0.001 |
| | UK | | Younger more likely to agree p<0.001 | Younger more likely to agree p<0.001 | Younger more likely to agree p<0.001 |
| **Gender** | Austria | n/s | n/s | n/s | Males more likely to agree (p<0.001) |
| | Denmark | | | Females more likely to agree than males (p = 0.030) | n/s |
| | UK | Males less likely to agree than females (p = 0.035) | | n/s | Males more likely to agree (p = 0.001) |
| **Live alone or not** | Austria | n/s | n/s | n/s | n/s |
| | Denmark | | | n/s | n/s |
| | UK | | | Live alone = more likely to agree (p = 0.036) | Live alone = more likely to agree (p = 0.008) |
| **Income** | Austria | n/s | n/s | n/s | n/s |
| | Denmark | Low (p = 0.040) and middle (p = 0.006) more likely to agree cf high | n/s | n/s | n/s |
| | UK | Middle (p = 0.016) less likely to agree cf high | No info (p = 0.019), low (p = 0.016) middle (p<0.001) less likely to agree cf high | Middle less likely to agree (p = 0.004) | Low (p = 0.042) and middle (p<0.001) were less likely to agree cf high |
| **Insurance** | Austria | n/s | Yes–more likely to agree (p<0.001) | Yes–more likely to agree (p = 0.001) | Yes–more likely to agree (p<0.001) |
| | Denmark | Yes = more likely to agree p = 0.002 | n/s | n/s | n/s |
| | UK | n/s | n/s | n/s | n/s |
| **Species** | Austria | n/s | n/s | n/s | Dog owners less likely to agree (p = 0.029) |
| | Denmark | | | | n/s |
| | UK | | | | n/s |
| **LAPS** | Austria | Higher LAPS = more likely to agree p<0.001 | Higher LAPS = more likely to agree p<0.001 | Higher LAPS = more likely to agree p = 0.012 | Lower LAPS = more likely to agree (p<0.001) |
| | Denmark | | | n/s | Lower LAPS = more likely to agree (p<0.001) |
| | UK | | | n/s | n/s |

0.000; $p_{AT,DK,UK} < 0.001$) as did owners with pet insurance compared to those without ($p_{ALL}$, $_{DK} < 0.001$; $p_{UK} < 0.008$; $p_{AT} = 0.032$). No significant relationship was found between owner age and LAPS score. In the pooled (all) data, owners who lived alone had significantly higher LAPS scores than those who lived with others ($p < 0.001$), however when countries were assessed individually, this effect was only apparent for Danish owners ($p < 0.001$), but not those in the UK or Austria. In regard to income, Danish and Austrian owners in the lower income

bracket had significantly higher LAPS scores than those in the high ($p_{AT,DK}$ = 0.000) or middle-income brackets ($p_{AT}$ = 0.013; $p_{DK}$ = 0.001), but this was not a significant factor for UK owners.

## Discussion

This paper presents results from a transnational study of dog and cat owners' expectations and attitudes towards modern AVC, contributing to the debate on whether modern veterinary care is 'fit for purpose', and the question of whether dogs and cats should have access to the same level of healthcare as human patients. Overall, 50% of the pet owners in our study agreed that their pet should be able to access the same *diagnostic tests* as human patients, while nearly 60% thought they should have access to the same *treatment options*. Owners were more likely to *disagree* with the statement that advanced care is 'unnecessary', but to be *neutral* on the question of whether advanced care has 'gone too far'. In all three countries, and judged by the highly significant p-values, the factor that was most strongly associated with the attitude that their pet should have access to the same treatment options as human patients was the level of attachment owners had to their pet (measured by using the LAPS).

Differences were noted in the attitudes of owners from the UK, Austria and Denmark, potentially influenced by the differences in the veterinary sector in each country. The majority of practices in Austria are small and privately owned (86% have less than 5 vets), larger practices are more common in Denmark (24% have 6–10 vets and 20% have 11–30 vets) and the UK (9% have 6–10 vets and 29% have 11–30 vets), with large corporate practices increasingly dominating in the UK [5, 10]. Our results reflect this, with the majority of dog and cat owners in Austria (69–71.7%) attending small (1–3 vet) practices, as do 57.9–58.6% of Danish owners, whereas a significantly higher proportion of UK owners (32.5%) attend practices with four or more vets. Although we did not specifically define 'advanced' veterinary care, the way in which owners selected the facilities / equipment that they would expect to be available at their usual practice suggested a common understanding across the three countries. Surprisingly however, half of the owners surveyed expected advanced options such as endoscopy, arthroscopy, CT and/or MRI to be available. Such equipment is normally only available at larger practices or Universities, as installation and running costs are high (e.g. MRI is £600-750K to purchase with annual service costs of £60K), and few primary care veterinarians have access to or experience in performing endoscopy or arthroscopy. Thus, pet owners seem to have higher expectations of their primary care veterinarians than of their primary care doctors, who would normally have to refer patients elsewhere for (basic) procedures such as dentistry, radiography or ultrasound. The between-country differences were significant and may reflect differences in practice sizes and relative stage of development of the profession in each country, as well as the degree of urbanization / proximity of owners to a veterinary practice. Approximately 60% of UK owners expected advanced diagnostic and treatment options to be available at their usual practice, in line with the fact that the UK currently has more large and specialised practices and a far greater number of veterinary specialists than either Denmark or Austria [10]. Austrian owners had similar expectations to UK owners despite being more likely to attend smaller independent practices, and it will be interesting to see whether this will lead to a change in the practice landscape in Austria in the future in the same way as has happened in the UK. In contrast, Danish owners were significantly less likely to expect advanced options to be available at their usual practices compared to UK and Austrian owners, potentially reflecting the lower level of attachment Danish owners have to their companion dogs and cats, identified in a separate study [37].

If a primary care veterinarian does not have the experience or equipment to treat a particular case, they are expected to offer the owner a referral to a more experienced or specialist

colleague who can offer the necessary (advanced) care. In our study, approximately 1 in 10 cat owners, and 1 in 5 dog owners have taken their pet(s) to a specialist, with the bias towards dogs consistent across the three countries. The overall low numbers likely reflect the fact that most medical problems can be managed in primary care practice. However, referral for advanced care may not always be an option, either because it is unaffordable for the owner (AVC is usually expensive), not available, or difficult to access, e.g. to those owners living in more isolated rural areas. The fact that more dogs than cats had been taken to a specialist reflects the situation in general primary care practice, where more dogs than cats are usually taken to the vet [38, 39]. There are many potential explanations for this, including the fact that cats tend to show clinical signs of illness less overtly than dogs as it is their nature to hide weakness or injury [40], they generally interact less frequently with their owners (who often consider them to be 'low maintenance'), and while most dogs enjoy travelling and exploring novel environments, transporting cats e.g. to the vets, is more challenging for all parties [39]. Another explanation may be the different levels of attachment that owners have towards dogs and cats, as the majority of studies show that dog owners are more strongly attached to their dogs than cat owners are to their cats [16, 41, 42], which may influence their willingness to seek veterinary care for them. A recent paper by our group [37] based on data from the same survey as the present paper found that in general cat owners were less willing to spend larger sums of money on veterinary care for their cats than dog owners were willing to spend on their dogs, although major cross-country differences were noted.

With respect to owners' attachment to their animal, studies have reported that in general a strong emotional bond with a pet can positively impact an owner's willingness to accept costly or prolonged veterinary treatment. For example, a strong emotional bond with a pet makes it more likely that an owner will agree to costly or prolonged veterinary treatment [19, 20], and some owners will even prioritise caring for their pets over their own healthcare needs, for example by delaying medical treatments or hospitalisation if pet care cannot be organised [21, 22]. This is reflected in our study, where owners who were more attached to their pets (higher LAPS scores) in all three countries were significantly more likely to believe that their pets should have access to the same treatment options as humans. However, owner age or living situation did not seem to be consistently associated with an expectation that their pet should have access to AVC, despite other studies having shown that the increasing numbers of single people, couples without children, and elderly people living alone tend to form particularly strong attachments to their pets [9]. Our results could be explained by the fact that we did not find a significant relationship between age and level of attachment (LAPS), and although overall those living alone were more attached to their pets, when comparing countries, this was significant only in the case of Danish owners, and not those in the UK or Austria. We did however find that female owners had significantly higher LAPS scores than male owners in all three countries in our study, in agreement with other studies reporting that females are significantly more attached to pets than males both as children [43], and as adults [20].

The fact that only half of the owners surveyed thought that pets should have access to the *same diagnostic* tests as human patients highlights an interesting distinction, as concerns have been raised in other studies that vets do too many tests [2, 8], even when motivated honorably [14]. An alternative explanation is that owners do not see the immediate or long-lasting benefits of undertaking tests, compared to undertaking a treatment that has a good outcome. Differences noted between countries could also reflect cultural differences, as discussed in a companion paper on owner preferences between dogs and cats in the same three countries [37] or simply potential ambiguity or a different interpretation of the question, as it did not explicitly state offering the most advanced treatment *along with other treatment options*. As discussed by others [44], most veterinarians in their study believed that all the options (not

just advanced diagnostics and treatments) should be offered to all clients, to avoid prejudging owner's preferences and to enable them to make a properly informed choice. Clients in that study seem to agree, reporting that when veterinarians only offered a single treatment option, they felt that it was more likely to be financially motivated [44].

As discussed in the introduction, while the advantages of access to AVC and specialists are obvious, there are also potential negative consequences for both the owner and the animal. For example, a strong emotional attachment to a pet can have adverse consequences if it causes owners to insist on pursuing treatments where the potential benefits are overemphasized and pain and suffering underestimated, leading to poor patient welfare, as owners, and vets, "try everything" [27]. In this context, it has been suggested that increasing specialization, pet insurance, undergraduate training, an increasingly litigious client base and widely publicised heroic treatments have pushed the profession in a direction of "gold standard" care, where it is easy for animal welfare to become secondary to other concerns [27]. Yet until now, opinions on AVC in this context have not been sought directly from pet owners themselves. Overall, owners in our study were more likely to *disagree* with the statement that advanced care is 'unnecessary' (as animals should not be treated as humans), but to be *neutral* on the question of whether advanced care has 'gone too far' (putting animals through too much). This may indicate, as suggested by others [45], that while owners may wish to pursue lifesaving treatment it will not be 'at all costs', as they will balance the quality and quantity of life achieved against the expense and potential for suffering.

Differences were evident between the countries however, as owners in Austria and Denmark who were less attached (lower LAPS scores) to their pets were significantly more likely to take the attitude that advanced care was unnecessary, but strength of attachment was not a factor in UK owners, perhaps because the majority expected and were used to such care being available in their usual practice. That Danish owners were more likely to be neutral on both questions may reflect a general lower attachment (LAPS) to their companion animals, and in particular to their cats [37], or simply a more pragmatic approach, for example favouring euthanasia over continued treatment of very sick animals if the prognosis is potentially guarded or poor. This perspective has been reported by others [45], who found that dog owners were more likely to choose euthanasia over treatment as their pet aged, and suggested that the trade-off between expensive veterinary care and gaining time with a pet that was beginning to decline becomes less favourable.

Although overall owners with pet insurance had a higher LAPS score than those without, neither income nor possession of pet insurance consistently predicted owner attitudes towards AVC across the three countries, except in Austria, where possession of insurance was significant, but income was not. The relationship between income and insurance was also not predictable across the three countries: we found no significant relationship between income and possession of insurance for Austrian and Danish owners, and in the UK, it was the high-income owners in the UK who were significantly more likely to have pet insurance than low-income owners. However further analysis of our data showed that in Denmark and Austria, owners in the lower income bracket had significantly higher LAPS scores than those in the high or middle-income brackets, and it may be that these owners would nonetheless be prepared to pay for AVC for their pets, irrespective of their financial situation.

That neither income nor possession of pet insurance consistently predicted owner attitudes towards AVC is perhaps surprising, in view of the increasing concerns over the affordability of veterinary care being highlighted in both the veterinary and popular press. For example, the Association of British Insurers recently reported that UK pet insurance payouts topped £1 billion for the first time in 2022 [46] and increasingly, owners are seeking help, such as crowd-funding, to pay vet bills [47]. Our results reflect those of other studies that have shown that as

the human–animal bond increases, pet expenditures also increase, including the willingness to select higher-cost treatments [45, 48–50]. This may offer some reassurance to the veterinary profession in the face of increasing concerns about the cost of veterinary care and sustainability of the pet insurance market, particularly during the current cost of living crisis [14, 46].

Finally, we explored whether owners were willing to help to advance veterinary care by agreeing to enrol their own pets into a research study. Although the majority of owners expected veterinarians to contribute to the advancement of veterinary care, they were less willing to enrol their own pet into a study to support this, with Danish owners being most reluctant. Interestingly, owners who were more attached to their pets (higher LAPS scores) were more likely to agree to enrol the pet into a research study, in all three countries. In asking about research, we did not make a distinction between 'basic' research, which is primarily aimed at understanding fundamental biological or disease processes without necessarily having a direct practical application, and clinical research, which generally has a direct application to improving patient care. Owners may therefore have prioritised the potential for harm to their pet over the likelihood of any direct benefits to future patients. Although veterinary clinical research directly benefits the animal species, such research on animal patients can also impact human health. The so-called 'One Health' approach recognizes the close interactions between animal and human health, and the potential benefits of clinical and research collaborations–see for example: https://www.cancer.gov/news-events/cancer-currents-blog/2019/comparative-oncology-dogs-cancer-clinical-trials. It seems likely that veterinarians will play an increasing role in 'One Health' studies, and recruitment of their animal patients into research studies will underpin this. While owner attitudes in this context merit further exploration, it will be useful for veterinarians recruiting to studies to know that younger owners were significantly more likely to agree to such a request than older owners, but that owner gender, living situation (alone or not), or pet species were not consistent factors in level of agreement in our study.

## Conclusions

Although just over half of the owners in our study believed that their pets should be able to access the same diagnostic tests and treatment as people, a significant proportion did not. While owners were more likely to disagree with the statement that advanced care for their pet is 'unnecessary', they were more neutral on the question of whether advanced care has 'gone too far', which merits further exploration. Not surprisingly, the factor that was most strongly associated with the attitude that their pet should have access to the same treatment options as human patients was the level of emotional attachment owners had to their pet (LAPS score), rather than factors such as owner age, living situation or finances. Further research exploring other factors contributing to owner decision-making within the context of advanced veterinary care is merited.

## Limitations

The questions posed were hypothetical, and owners may have expressed different opinions or taken different actions in a real-life scenario in which they actually had to make treatment decisions about a sick pet of their own. We did not ask respondents about their experience of caring for sick pets, or veterinary care, which is likely to influence attitudes to AVC. Decision-making may also be influenced by other factors not tested here, such as the age of the animal or the relationship with their veterinarian. Although the statements and questions included in the study were pre-tested, some may have remained ambiguous and therefore open to being interpreted differently by individuals, for example, we did not specifically define 'advanced' care, or 'specialist' nor did we check respondents understanding of the terms.

Internet panels were used to collect data, and the recruitment principle in internet panels does not follow the classical probability-based sampling principle where each unit of the target population has an equal chance of being drawn [51]. Recent research has shown that findings from internet panels deviate from data collections that follow the gold standard of probability-based sampling. When probability-based recruitment to the panel is employed, the average absolute error rate is 2.6%; when non-probability recruitment is used, it increases to 5.8% [52]. When it comes to pet owners, we do not know the exact error rate, because this topic was not covered [52] but we can of course expect some level of error. In Denmark, where some panel members were invited using a random procedure (telephone with random digit dialing), the error in reported proportions and means is likely to be smaller than in the UK and Austria where a probability-based procedure was not used in the recruitment of panel members.

Lastly, the survey was limited to owners in three (wealthy) European countries, and results may have differed had less affluent socioeconomic countries or regions been included, particularly areas where a good standard of human healthcare is not accessible to all.

## Supporting information

**S1 File. Questionnaire.**
(DOCX)

**S2 File. Ordinal regression analysis of socio-demographic and financial factors and LAPS scores, on attitudes towards advanced veterinary care and research.**
(DOCX)

**S3 File. Type of practice usually attended by owners of dog(s) and / or cat(s).**
(DOCX)

**S4 File. Pearson Correlation analysis to explore whether owners who believe their pets should have access to advanced veterinary care are also likely to think that their vets should contribute knowledge to the advancement of such care and be willing to enrol their own pets into a research study.**
(DOCX)

**S5 File. Differences in uptake of pet health insurance and income.**
(DOCX)

**S6 File. Correlations between LAPS scores and age, gender, living status, insurance and income.**
(DOCX)

## Acknowledgments

The authors wish to thank dog and cat owners who participated in this study, and the colleagues and owners who helped to refine the questionnaire by participating in the cognitive interviews and pilot study.

## Author Contributions

**Conceptualization:** Sandra A. Corr, Thomas Bøker Lund, Peter Sandøe, Svenja Springer.

**Data curation:** Svenja Springer.

**Formal analysis:** Svenja Springer.

**Funding acquisition:** Peter Sandøe.

**Investigation:** Sandra A. Corr, Thomas Bøker Lund, Svenja Springer.

**Methodology:** Sandra A. Corr, Thomas Bøker Lund, Peter Sandøe, Svenja Springer.

**Project administration:** Sandra A. Corr, Thomas Bøker Lund, Peter Sandøe, Svenja Springer.

**Resources:** Peter Sandøe.

**Visualization:** Sandra A. Corr.

**Writing – original draft:** Sandra A. Corr.

**Writing – review & editing:** Sandra A. Corr, Thomas Bøker Lund, Peter Sandøe, Svenja Springer.

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
