## [Decision Letter · Decision Letter 0]

30 Nov 2023

PONE-D-23-28989Cat and Dog Owners’ Expectations and Attitudes towards Advanced Veterinary Care (AVC) in the UK, Austria and Denmark.PLOS ONE

Dear Dr. Springer,

Thank you for submitting your manuscript to PLOS ONE. After careful consideration, we feel that it has merit but does not fully meet PLOS ONE’s publication criteria as it currently stands. Therefore, we invite you to submit a revised version of the manuscript that addresses the points raised during the review process.

We look forward to receiving your revised manuscript.

Kind regards,

Francesca Baratta, PharmD, PhD

Academic Editor

PLOS ONE

Reviewers' comments:

Reviewer's Responses to Questions

**Comments to the Author**

1. Is the manuscript technically sound, and do the data support the conclusions?

Reviewer #1: Yes

Reviewer #2: Yes

2. Has the statistical analysis been performed appropriately and rigorously? 

Reviewer #1: I Don't Know

Reviewer #2: Yes

3. Have the authors made all data underlying the findings in their manuscript fully available?

Reviewer #1: Yes

Reviewer #2: Yes

4. Is the manuscript presented in an intelligible fashion and written in standard English?

Reviewer #1: Yes

Reviewer #2: Yes

5. Review Comments to the Author

Reviewer #1: In this questionnaire survey study, sent to cat and dog owners, representative samples, in the UK, Austria and Denmark, the authors investigated owners’ expectations and attitudes towards advanced veterinary care. Also, the factors that might influence those views were analysed.

In line with arguments presented within the veterinary community during the resent years, it was speculated there may be negative consequences for the well-being of the animals: if euthanasia actually is appropriate, owners with strong emotional attachments to their pets may pursue treatments that significantly reduce the quality of the animal’s life, with the aim to prolong their pet’s life. This contributes to an ethical discussion, what is right, or wrong, is the customer always right? Another aspect is the concept of ‘caregiver burden’ for the pet owner, emotional distress, financial and practical challenges that comes with AVC. It was investigated if this was correlated to level of emotional attachment.

Furthermore, it was studied if owners who expect AVC to be available for their animal were more likely to be willing to participate in a clinical research study.

The authors reported 59% of the pet owners believed that their pets should have access to the same treatment options as humans. This is indeed an interesting result, and I reach for a clarification: what does “access” mean? Should be given (moving towards “avoid euthanasia at all cost”)? Should be available, ie the treatment option should simply be there, financial aspects to be considered / taken into consideration etc?

These initial comments are along the path of “are the questions to a high or low degree open to interpretation”? However, if there was room for interpretation, this should have been caught in the test run, which was appropriately performed. Also, this is covered in section Limitations, “some may have remained ambiguous and therefore open to being interpreted differently by individuals”.

The comments are within Minor revision, mainly related to style preferences.

The only Major revision is how to deal with cultural differences between countries, and inclusion/exclusion criteria (Should respondents who work within the veterinary field (6.8%) be excluded?). Please see below.

Overall, this is a rather unique study, important not only for vets, also for our health care industry, including insurance companies. Has modern veterinary care “gone too far”? Opinions on AVC on this topic have not been sought directly from pet owners themselves. In addition, some very interesting cultural differences between countries were found, also differences between genders. Publication of this manuscript is clearly justified.

Ethics: was there only a permit for the Kingdom of Denmark, not the other countries? If so, is this appropriate?

Abstract

Line 55-57: please consider if the wording is appropriate, neutral and disagree.

Line 63: Our findings will help inform veterinarians… not only veterinarians; also other health care providers, and others related to our industry (insurance companies).

Introduction

Line 132: veterinarian salaries… suggest “staff salaries”, as there are several professions involved.

Line 144: suggest “was the focus”

Line 146-158: Please consider significantly shorten this section; parts of the text may be more suitable in the Discussion.

Line 181: Suggest “This study addressed”

Line 193: Suggest “were explored”

Line 198-202: Please consider moving to first section Discussion; suggest include insurance companies.

M&M

Line 206: can the Danes give permission for a study performed in other countries? Perhaps the answer is yes, taken into consideration where the Norstat company is based?

Results

Should respondents who work within the veterinary field (6.8%) be excluded? Can they be considered “representative pet owners”?

Line 425-426: Please consider the appropriate wording – does the text accurately describe the difference between 45% (neutral) and 40% (disagree)? …neutral on the question of whether advanced care has ‘gone too far’ (45.3%), and tended to disagree with the statement that it is ‘unnecessary’ (40.1%).

Discussion

Line 532: no need to repeat results – please consider removing p-values.

Line 555: There are several possible reasons for the between-country differences; apart from size of clinics, other cultural differences may play a large role (what attitudes towards animals are we transferring to our children?). A clue is found in the abundance of pets; UK pet owners often have several more pets per household than citizens of other countries do. Does a higher proportion of households in the UK have pets compared to other countries? My message is that clearly there are differences between countries in attitudes, and this study nicely demonstrates so. The question is, are you overemphasizing the importance of size (smiley)? Isn’t there is a lot more behind those differences than mean size of practise? Please consider broadening the text. This may also resolve line 559-60 “which is difficult to explain”.

The crosscheck study would be to interview vets in the UK (or other nation) that originate from another country/culture and have working experience from that other country. What attitudes amongst our customers do we recognise between countries, and what attitudes differ? A potential topic for another study…

Line 570-576: can the topic of “culture and attitude” also be incorporated? It would seem dogs are generally considered more valuable than cats in several countries, as dogs are more likely to be insured… and owners are in general more willing to spend £ on their dogs (than cats). This might be a poor parallel to societal hierarchy, but in general, the MD and CEO are considered “more important” than another random profession; is this part of the explanation, dogs are simply more valuable (where attachment is only one factor)?

Line 606-609: “Differences noted between countries could also reflect cultural differences”. Thank you!

Reviewer #2: A very interesting manuscript with results that will be valuable for those working in veterinary medicine.

Abstract

Line 50 – ‘…questionnaire survey…’ – suggest changing to ‘…questionnaire used to survey cat and dog owners…’

Introduction

-There is information about the benefits of AVC, but is there any data suggesting reduced success rates in pets compared to humans for certain types of AVC? Or potential health, behavior, other, consequences for pets?

Line 99 – what exactly is in the ‘interest’ of the pet? Instead, suggest adding a sentence with details on the potential animal welfare implications, and providing more examples to back up the claim that AVC may not be in the best ‘interest’ of pets.

Line 110 – I don’t think veterinarians are highly motivated to use all new technologies (example virtual care via telemedicine), but I think you’re referring to specific types of technologies here… can you be more specific?

Lines 116-119 don’t seem relevant, suggest taking out

Line 146 – can you add ‘We predict…’ at the beginning of this sentence? It sounds more like a statement the way it’s currently written

Line 182 – wouldn’t the second part of research question 1 depend on whether a pet owner has had a pet with a serious health issue or was given the option/ recommendation to see a specialist? Did you screen for this first, then see the number/percentage who chose to see a specialist of those that have had a pet with a serious health issue that wasn’t being managed by their primary veterinarian?

Line 198 – suggest changing ‘…our study will help to give…’ to ‘… our study aims to provide…’

Materials and Methods

Line 213 – is there evidence that NORSTAT can reach a representative sample? In other words, are pet owners that choose to participate in NORSTAT surveys, different than those that choose not to participate? Is there any data/research on this?

Line 214 – states ‘…., including pet owners.’ Did your survey include non-pet owners? If so, why? This part is confusing because in the introduction you specify your target population is pet owners.

Line 215 – What was the randomization procedure? For example, a random number generator. Please specify.

Line 215 – How do you know those from NORSTAT’s pre-established citizen panel is representative of the average pet owner?

Was there an incentive (i.e., monetary) to participate?

Line 227 – 230 about the consent process is confusing/wordy, suggest being more concise. For example, participants provided consent by clicking a ‘next’ button at the bottom of the consent form to proceed to the questionnaire.

Line 260 – Survey design and measurement – can you move this section up before recruitment of participants? The organization of the methods isn’t in order of how the study was conducted, which is a bit confusing. Suggest re-organizing the sections of the Materials and Methods to be more sequential, which I think may help with clarity/understanding. For example, order of sub-sections could be: Ethics, survey design and measurement, survey development, recruitment

Line 274 – how does asking them to chose ‘their favorites pet’ bias the results? Why not ask them to choose the pet whose name comes first in the alphabet? Also, do you know if pet owners with both dogs and cats, commonly chose to answer questions for their dog versus cat? I would think answering for their favorite pet would be the pet they're more likely to spend money on.

Data Analyses

Line 321 – for the ‘animal species’ categorical variables in the models, did you not also have a ‘both cat and dog owner’ category? Or for these models did you only include cat owner, and dog owner, not those who are both?

Line 322 – Why include ‘prefer not to say/I don’t know’ option for the total gross household income per year variable? What does this information provide?

Line 328-329 – Ok now I understand! Suggest moving this information up so it’s clear in your methods section. Also suggest moving this up when you first talk about that explanatory variable in the data analyses section. Can you also provide rationale for choosing to do this?

Overall there is no information about how you assessed model fit, or tested model assumptions. Please add this.

Results

Line 250 -chi-square tests are inferential tests, and not descriptive. Suggest changing the title of this section.

Discussion

Line 583 – Could another explanation be that dog owners are more likely to spend more money on their pets than cat owners?

Overall, a great job summarizing the results and incorporating with current literature.

Tables/Figures/Supporting Information:

-Please provide a more descriptive caption for tables, as the reader should be able to interpret the tables without having to read the manuscript. For example, please add ‘N’, outcome variable categories, population, etc.

6. PLOS authors have the option to publish the peer review history of their article (what does this mean?). If published, this will include your full peer review and any attached files.

Reviewer #1: No

Reviewer #2: No

---

## [Author Response · Author response to Decision Letter 0]

16 Jan 2024

Response to Reviewers

Reviewer #1: In this questionnaire survey study, sent to cat and dog owners, representative samples, in the UK, Austria and Denmark, the authors investigated owners’ expectations and attitudes towards advanced veterinary care. Also, the factors that might influence those views were analysed.

In line with arguments presented within the veterinary community during the resent years, it was speculated there may be negative consequences for the well-being of the animals: if euthanasia actually is appropriate, owners with strong emotional attachments to their pets may pursue treatments that significantly reduce the quality of the animal’s life, with the aim to prolong their pet’s life. This contributes to an ethical discussion, what is right, or wrong, is the customer always right? Another aspect is the concept of ‘caregiver burden’ for the pet owner, emotional distress, financial and practical challenges that comes with AVC. It was investigated if this was correlated to level of emotional attachment.

Furthermore, it was studied if owners who expect AVC to be available for their animal were more likely to be willing to participate in a clinical research study.

The authors reported 59% of the pet owners believed that their pets should have access to the same treatment options as humans. This is indeed an interesting result, and I reach for a clarification: what does “access” mean? Should be given (moving towards “avoid euthanasia at all cost”)? Should be available, ie the treatment option should simply be there, financial aspects to be considered / taken into consideration etc?

The reviewer raises an interesting point, and in the context of the manuscript, we use ‘access’ to mean ‘to be available’, rather than whether it should be given, which we explore in the questions on whether advanced care is unnecessary, or has gone too far. However, as the reviewer notes below, this did not seem to cause ambiguity in the pilot trials.

These initial comments are along the path of “are the questions to a high or low degree open to interpretation”? However, if there was room for interpretation, this should have been caught in the test run, which was appropriately performed. Also, this is covered in section Limitations, “some may have remained ambiguous and therefore open to being interpreted differently by individuals”. The comments are within Minor revision, mainly related to style preferences.

The only Major revision is how to deal with cultural differences between countries, and inclusion/exclusion criteria (Should respondents who work within the veterinary field (6.8%) be excluded?). Please see our comments below.

Overall, this is a rather unique study, important not only for vets, also for our health care industry, including insurance companies. Has modern veterinary care “gone too far”? Opinions on AVC on this topic have not been sought directly from pet owners themselves. In addition, some very interesting cultural differences between countries were found, also differences between genders. Publication of this manuscript is clearly justified. Thank you.

Ethics: was there only a permit for the Kingdom of Denmark, not the other countries? If so, is this appropriate? Yes, we only applied for approval at the IRB (Ethics Committee) of the Science Faculty at University of Copenhagen (UCPH) that manages and organizes the project. 

Data collection from all three countries was handled by the survey company, and this company is the only institution that has access to person identifiable information of the respondents. The handling of this person identifiable data was agreed upon with the UCPH team according to the European GDPR rules. Further, the company handling the data was evaluated and approved by the GDPR department at the Science faculty of UCPH. Following this GDPR approval the ethics committee at Science, UCPH approved the project. 

We believe that it is the standard and appropriate way to ask for approval when the research and data collection is organized and managed from one institution.

Abstract

Line 55-57: please consider if the wording is appropriate, neutral and disagree. 

We have changed the wording to ‘ Owners were most likely to be neutral on the question of whether advanced veterinary care has ‘gone too far’ (45.3%), and to disagree with the statement that that advanced care is ‘unnecessary’ (40.1%). Lines 55-57.

Line 63: Our findings will help inform veterinarians… not only veterinarians; also other health care providers, and others related to our industry (insurance companies). This is a good point – we have changed to ‘and other health care providers’ – line 63. Further, we have expanded the point at the end of the introduction - lines 199-201.

Introduction

Line 132: veterinarian salaries… suggest “staff salaries”, as there are several professions involved. This is a good point and we have changed as suggested – line 137.

Line 144: suggest “was the focus” Changed as suggested – line 149. 

Line 146-158: Please consider significantly shorten this section; parts of the text may be more suitable in the Discussion. We have moved some of the text from here to the discussion – lines 610-614.

Line 181: Suggest “This study addressed”. Changed as suggested – line 183.

Line 193: Suggest “were explored”. Changed as suggested – line 195.

Line 198-202: Please consider moving to first section Discussion; The second part of this paragraph has been shortened and moved to the beginning of the discussion – lines 543-546. 

 Suggest include insurance companies. Insurance companies included – line 200.

M&M

Line 206: can the Danes give permission for a study performed in other countries? Perhaps the answer is yes, taken into consideration where the Norstat company is based? Please see comment addressing this in response to ‘Ethics’ above. 

Results

Should respondents who work within the veterinary field (6.8%) be excluded? Can they be considered “representative pet owners”? We believe it is appropriate to include those who work within the veterinary field. We are trying to assess prevalence and patterns among pet owners in general, and workers in the veterinary field are certainly also a segment of pet owners. So, it would make the results less representative if we removed these observations. 

Line 425-426: Please consider the appropriate wording – does the text accurately describe the difference between 45% (neutral) and 40% (disagree)? …neutral on the question of whether advanced care has ‘gone too far’ (45.3%), and tended to disagree with the statement that it is ‘unnecessary’ (40.1%). The wording has been modified and now reads ‘ Owners were most likely to be neutral on the question of whether advanced care has ‘gone too far’ (45.3%), and to disagree with the statement that it is ‘unnecessary’ (40.1%) - Lines 440-442 .This has also been changed in the abstract – line 55-57.

Discussion

Line 532: no need to repeat results – please consider removing p-values. All p-values have been removed from the discussion.

Line 555: There are several possible reasons for the between-country differences; apart from size of clinics, other cultural differences may play a large role (what attitudes towards animals are we transferring to our children?). A clue is found in the abundance of pets; UK pet owners often have several more pets per household than citizens of other countries do. Does a higher proportion of households in the UK have pets compared to other countries? My message is that clearly there are differences between countries in attitudes, and this study nicely demonstrates so. The question is, are you overemphasizing the importance of size (smiley)? Isn’t there is a lot more behind those differences than mean size of practise? Please consider broadening the text. This may also resolve line 559-60 “which is difficult to explain”. The reviewer raises a good point, and we have modified and broadened the text as suggested – lines 572-585. 

The crosscheck study would be to interview vets in the UK (or other nation) that originate from another country/culture and have working experience from that other country. What attitudes amongst our customers do we recognise between countries, and what attitudes differ? A potential topic for another study…We agree that would be an interesting future study.

Line 570-576: can the topic of “culture and attitude” also be incorporated? It would seem dogs are generally considered more valuable than cats in several countries, as dogs are more likely to be insured… and owners are in general more willing to spend £ on their dogs (than cats). This might be a poor parallel to societal hierarchy, but in general, the MD and CEO are considered “more important” than another random profession; is this part of the explanation, dogs are simply more valuable (where attachment is only one factor)? We have added a comment referencing a paper recently published by our group that expands on this – lines 582-585.

Line 606-609: “Differences noted between countries could also reflect cultural differences”. Thank you! You are welcome!

Reviewer #2: A very interesting manuscript with results that will be valuable for those working in veterinary medicine. Thank you.

Abstract

Line 50 – ‘…questionnaire survey…’ – suggest changing to ‘…questionnaire used to survey cat and dog owners…’ We have changed the sentence as suggested – line 50-51

Introduction

-There is information about the benefits of AVC, but is there any data suggesting reduced success rates in pets compared to humans for certain types of AVC? Or potential health, behavior, other, consequences for pets? A lot has been published on outcomes for specific advanced treatments in animals, and comparable outcomes for humans are referenced – for example survival times are improved in both humans and animals following chemotherapy for chemo-responsive tumours. However, the lifespans and behavioural needs are very different between animals and humans of course, making such direct comparison of limited validity. In more general terms, we address the question of whether AVC can negatively impact animals if for example quantity of life is prioritised over quality of life – see for example lines 100-102, and 162-164.

Line 99 – what exactly is in the ‘interest’ of the pet? Instead, suggest adding a sentence with details on the potential animal welfare implications, and providing more examples to back up the claim that AVC may not be in the best ‘interest’ of pets. We have reworded this as follows: As a broader principle, concern has also been raised over whether the prolongation of life (animal or human) at all costs is necessarily always in line with what may be considered the best interests of the patient, especially when the (animal) patients cannot express their own interests [2,4]. For example, some may argue that it is not in a dog’s best interest to undergo chemotherapy involving repeated veterinary visits and hospital stays, and associated malaise, to gain an extra 6 months with its owner. Lines 97-102.

Line 110 – I don’t think veterinarians are highly motivated to use all new technologies (example virtual care via telemedicine), but I think you’re referring to specific types of technologies here… can you be more specific? The study we refer to did not detail specific technologies by name, but rather in principle, and so we have reworded this to better reflect the results of that study – lines 111-115.

Lines 116-119 don’t seem relevant, suggest taking out. We believe the information on advanced training is worth noting, as it underpins the increasing availability of - and potentially drive to offer- advanced care, which is the subject of the paper. Lines 120-124.

Line 146 – can you add ‘We predict…’ at the beginning of this sentence? It sounds more like a statement the way it’s currently written. ‘We predict that’ has been added as suggested – line 151.

Line 182 – wouldn’t the second part of research question 1 depend on whether a pet owner has had a pet with a serious health issue or was given the option/ recommendation to see a specialist? Yes it would, but we think that is implicit in the question as you would not see a specialist if your pet did not need advanced care. 

Did you screen for this first, then see the number/percentage who chose to see a specialist of those that have had a pet with a serious health issue that wasn’t being managed by their primary veterinarian? No we did not screen for this i.e. ask how many pet owners had the option to take their pet to a specialist but had declined to do so. We hope that this is adequately addressed in the limitations section. 

Line 198 – suggest changing ‘…our study will help to give…’ to ‘… our study aims to provide…Changed as suggested- Line 199.

Materials and Methods

Line 213 – is there evidence that NORSTAT can reach a representative sample? In other words, are pet owners that choose to participate in NORSTAT surveys, different than those that choose not to participate? Is there any data/research on this? 

To our knowledge there is no study available to answer the specific question about whether internet panels such as NORSTAT can represent the population sub-group of “pet owners" accurately. At a broader level, though, there is a recent study that looked at the accuracy of which internet panels such as Norstat represent the general target population. This study suggests that online panels are relatively accurate if weights are used to account for over-representation of demographic strata, e.g. men relative to women compared to the population census. The level of error depends on the recruitment procedure of panel members to the panel. If non-probability methods are used, the average absolute error (AAE) is 5.6%, while, if non-probability recruitment is used the AAE is 2.5%*. In Denmark, participants are invited through a combination of digital media (typically invitation through internet pages) as well as random digit telephoning. While this does not completely follow the probability sampling principle that all units from a sample frame has an equal probability of being recruited, it does come quite close. On the other hand, in the UK and in Austria, Norstat does not use a similar probability-based principle, as, in these two countries, participants are only recruited through digital media. This means that the absolute AAE will be higher. 

* Mercer, A., & Lau, A. (2023). Comparing two types of online survey samples. Pew Research Center, USA. September 2023. Online at: https://www.pewresearch.org/methods/2023/09/07/comparing-two-types-of-online-survey-samples/

As mentioned on line 297-301, we used country-specific weight variables in all reports of means and proportions. So, in that sense, we do believe that the data will come relatively close to an accurate portrayal of pet owners. Still, having the description above about AAE in mind, we have now inserted a paragraph in the limitation section about this – lines 742-753.

Line 214 – states ‘…., including pet owners.’ Did your survey include non-pet owners? If so, why? This part is confusing because in the introduction you specify your target population is pet owners. There are two reasons that the survey included non-pet owners. 1) The paper is part of a larger study with several aims, including, for example, understanding why people do or don’t keep pets – hence the original sample contained non pet owners. 2) An appropriate way to obtain a representative sample of pet owners is to conduct a population-level sampling (consisting of both pet and non-pet owners), as the pet owners that can be singled out from the larger sample will then be representative of pet owners in the population. Indeed, to our knowledge it would not be possible to achieve a representative sample of pet owners through other routes than the population-level sampling that we used, as there is no way that we only can identify and recruit pet owners to partake in a survey. 

Line 215 – What was the randomization procedure? For example, a random number generator. Please specify.

For internet panels, there are two invitation processes. The first process is being invited to the panel (we described this in an earlier remark to your query at your line 213 question above). The second process is, once you are a member of the panel, being invited to questionnaire studies. For the second invitation process, the lottery principle that is used to invite panelists randomly is not based on commercial software (e.g. SAS) with a program for carrying out the randomization procedure. Neither is it based on freeware, e.g. random.org. Norstat uses their own program, which may vary across countries, and across the panels that Norstat co-operated with to collect data. Here is the code used in Denmark:

public static IList<T> Shuffle<T>(this IList<T> list)

{

var n = list.Count;

while (n > 1)

{

var k = RNG.Next(n);

n--;

var value = list[k];

list[k] = list[n];

list[n] = value;

}

return list;

}

Line 215 – How do you know those from NORSTAT’s pre-established citizen panel is representative of the average pet owner? As mentioned in an earlier response to you, we aimed to obtain samples drawn to be representative of the overall populations in UK, Austrian and Denmark. By implication, the pet owners that we can single out from these samples will be representative of cat and dog owners in the three countries. This reasoning is no different than if we on the basis of a sample that is representative of an entire population conduct sub-group analyses of e.g. prevalences and patterns of associations in a particular socio-demographic segment, e.g. people in the age groups 38-49 years. As further mentioned in an earlier response to you, we do not have census data through which we can benchmark whether the pet owner samples of the three countries are accurate regarding e.g. pet owner’s demographic profile. 

We thank the reviewer for these important queries about the data quality, and as noted above, have included a paragraph in the limitations section about possible sources of error stemming from the data.

Was there an incentive (i.e., monetary) to participate? NORSTAT panel members are rewarded in various ways for participating in surveys, including a choice of gift cards or digital coupons, and the option to donate their points to plant a tree, or to a charity – see https://www.norstatpanel.com/en

Line 227 – 230 about the consent process is confusing/wordy, suggest being more concise. For example, participants provided consent by clicking a ‘next’ button at the bottom of the consent form to proceed to the questionnaire. This has been made more concise as suggested – lines 287-289. 

Line 260 – Survey design and measurement – can you move this section up before recruitment of participants? The organization of the methods isn’t in order of how the study was conducted, which is a bit confusing. Suggest re-organizing the sections of the Materials and Methods to be more sequential, which I think may help with clarity/understanding. For example, order of sub-sections could be: Ethics, survey design and measurement, survey development, recruitment. This section has been reordered as suggested by the reviewer – lines 208-260.

Line 274 – how does asking them to chose ‘their favorites pet’ bias the results? Why not ask them to choose the pet whose name comes first in the alphabet? Also, do you know if pet owners with both dogs and cats, commonly chose to answer questions for their dog versus cat? I would think answering for their favorite pet would be the pet they're more likely to spend money on.

We asked them to choose their favourite pet for the reason the reviewer states – that they would be more likely to spend money on AVC their favourite pet. We believe that this will give a truer representation of how owners feel about AVC than asking them to think about treating a pet they didn’t care as much about. In addition, we were aware of the potential for survey fatigue – asking 23 questions for each family pet would clearly be too much. 

We have added more detailed information concerning the proportion of owners who chose cat or dog as their favourite pet; indicating that 65.6% chose dog and 26.7% chose cat as favourite pet. Lines 318-322. 

Data Analyses

Line 321 – for the ‘animal species’ categorical variables in the models, did you not also have a ‘both cat and dog owner’ category? Or for these models did you only include cat owner, and dog owner, not those who are both? See our actions regarding this in relation to your comment below.

Line 322 – Why include ‘prefer not to say/I don’t know’ option for the total gross household income per year variable? What does this information provide? From an ethical review perspective, a questionnaire must have an answer option that permits the respondent not to answer when personal information is being requested. If the question relates to why the variable was included in the data analysis, this was simply to avoid having to remove a relatively large number of pet owners that would/could not disclose their income from the data analysis

Line 328-329 – Ok now I understand! Suggest moving this information up so it’s clear in your methods section. Also suggest moving this up when you first talk about that explanatory variable in the data analyses section. This section has been moved as requested – now lines 304-312. 

Can you also provide rationale for choosing to do this? This has been addressed to some degree above. However the rationale is also that the paper focusses on a comparison of dogs and cats, and we wanted to keep it as “simple” and straight forward as possible; not only for the analysis, but also for the interpretation, discussion of data and conclusions. Against the background that less than 20% of owners stated they have both dog and cat, we have a good reason to create a binary variable which supports our main idea of the paper. Further, it is likely that owners will think about their favourite pet (dog or cat) while answering the statements related to modern small animal practice. This may also be triggered by our question about the favourite pet before answering the LAPS (which we also addressed before the statements related to modern small animal practice). In addition, we formulated all statements in singular (“My pet… ” and never “My pet(s)…” which may further stimulate respondents to think about one (the favourite) pet. 

Overall there is no information about how you assessed model fit, or tested model assumptions. Please add this. We have added the following to the data analysis section: For each regression analysis, we evaluated the overall regression model fit using the LR chi2 test and Nagelkerke’s Pseudo-R-Square (see Supporting Information 2). Lines 335-337. 

Results

Line 250 -chi-square tests are inferential tests, and not descriptive. Suggest changing the title of this section. Apologies – the title has been changed to “Dog and cat owner socio-demographics. Line 346.

Discussion

Line 583 – Could another explanation be that dog owners are more likely to spend more money on their pets than cat owners? Yes, we believe that is what we have explained and referenced – lines 604-607.

Overall, a great job summarizing the results and incorporating with current literature. Thank you! 

Tables/Figures/Supporting Information:

-Please provide a more descriptive caption for tables, as the reader should be able to interpret the tables without having to read the manuscript. For example, please add ‘N’, outcome variable categories, population, etc. We have done what we believe the reviewer is requesting for each table. 

6. PLOS authors have the option to publish the peer review history of their article (what does this mean?). If published, this will include your full peer review and any attached files.

Do you want your identity to be public for this peer review? For information about this choice, including consent withdrawal, please see our Privacy Policy.

Reviewer #1: No

Reviewer #2: No

---

## [Decision Letter · Decision Letter 1]

8 Feb 2024

Cat and Dog Owners’ Expectations and Attitudes towards Advanced Veterinary Care (AVC) in the UK, Austria and Denmark.

PONE-D-23-28989R1

Dear Dr. Corr,

We’re pleased to inform you that your manuscript has been judged scientifically suitable for publication and will be formally accepted for publication once it meets all outstanding technical requirements.

Kind regards,

Francesca Baratta, PharmD, PhD

Academic Editor

PLOS ONE

Reviewer's Responses to Questions

**Comments to the Author**

1. If the authors have adequately addressed your comments raised in a previous round of review and you feel that this manuscript is now acceptable for publication, you may indicate that here to bypass the “Comments to the Author” section, enter your conflict of interest statement in the “Confidential to Editor” section, and submit your "Accept" recommendation.

Reviewer #1: All comments have been addressed

Reviewer #2: All comments have been addressed

2. Is the manuscript technically sound, and do the data support the conclusions?

Reviewer #1: Yes

Reviewer #2: Yes

3. Has the statistical analysis been performed appropriately and rigorously? 

Reviewer #1: Yes

Reviewer #2: Yes

4. Have the authors made all data underlying the findings in their manuscript fully available?

Reviewer #1: Yes

Reviewer #2: Yes

5. Is the manuscript presented in an intelligible fashion and written in standard English?

Reviewer #1: Yes

Reviewer #2: Yes

6. Review Comments to the Author

Reviewer #1: The authors have commented and addressed the expressed concerns and suggestions appropriately.

This is a fine manuscript, well worthy of publication.

Reviewer #2: (No Response)

7. PLOS authors have the option to publish the peer review history of their article (what does this mean?). If published, this will include your full peer review and any attached files.

Reviewer #1: No

Reviewer #2: No

---

## [Editor Report · Acceptance letter]

27 Feb 2024

PONE-D-23-28989R1 

PLOS ONE

Dear Dr. Corr, 

I'm pleased to inform you that your manuscript has been deemed suitable for publication in PLOS ONE. Congratulations! Your manuscript is now being handed over to our production team.

Kind regards, 

on behalf of

Dr. Francesca Baratta 

Academic Editor

PLOS ONE